

# No-tillage with Stubble Mulching Enhances Soil Physical Properties and Reduces Soil Penetration Resistance: A Comparative Study in Mollisol Region of Northeast China

Dawei Wang[1], Hao Sun[2], Linding Wei[1], Boxiang Wang[1], Jinyou Qiao[1, 3, *], Jian Sun[1, 3, 4, *],

Haitao Chen[1, 3, 4, *]

[1]College of Engineering, Northeast Agricultural University, Harbin, 150030, Heilongjiang, China
[2]Aerospace Life-support Industries, Ltd, Xiangyang, 441003, Hubei, China
[3]Heilongjiang Province Technology Innovation Center of Mechanization and Materialization of Major Crops Production, Harbin, 150030, Heilongjiang, China
[4]College of Mechatronic Engineering, East University of Heilongjiang, Harbin, 150066, Heilongjiang, China.

**Corresponding author:**

E-mail addresses: jyqiao@neau.edu.cn (Jinyou Qiao), dnsunjian@163.com (Jian Sun), htchen@neau.edu.cn (Haitao Chen)

**Abstract.** The mollisol region of Northeast China constitutes a critical grain production base. However, prolonged intensive farming has disrupted native soil structures, driving soil degradation and generating excessive crop residues that constrain

sustainable agricultural development. To address these challenges, a field experiment evaluated four mechanized tillage-sowing practices: Plow Tillage with Precise Sowing (PTS), Rotary Tillage with Precise Sowing (RTS), No-Tillage Sowing (NTS), and No-Tillage with Stubble Mulching and Sowing (NTMS). This study systematically assessed the impacts of these practices on soil compaction through analysis of soil penetration resistance (SPR), while further examining their effects on soil water content (SWC) and soil bulk density (SBD). Results demonstrated that NTMS significantly increased SWC, whereas

NTS resulted in higher SBD and SPR than other practices. Both PTS and RTS improved SWC relative to NTS and reduced SBD more effectively than NTS or NTMS. Across all practices, SPR exhibited consistent trends during the soybean growth cycle, peaking at the podding stage. NTMS outperformed alternative practices by optimizing soil physical properties, thereby enhancing soil quality and slowing degradation processes in the black soil. Collectively, NTMS implemented within a maize-soybean rotation system offers a viable solution to address maize straw surplus and soil degradation in Northeast China's

mollisol region.

**Keywords.** mechanized tillage practices; soil physical properties; soil penetration resistance; Mollisol conservation

## 1 Introduction

As one of the three major mollisol regions globally, the Northeast China mollisol region spans 180,000 km² and constitutes China's most critical grain production base (Spaargaren and Deckers, 1998). This region accounts for over 30% of national

maize output and nearly 50% of soybean output (Wang et al., 2022; Liu et al., 2021; Ma et al., 2019; Wu et al., 2022). However,





expanding maize cultivation and rising crop yields have generated excessive straw residues while exacerbating soil degradation through intensive tillage. These issues threaten conventional farming sustainability.

Conservation tillage optimizes soil structure for crop growth by enhancing water retention, facilitating root development, and increasing yield potential. Relative to conventional tillage, it significantly improves soil water content (SWC) and organic matter accumulation, thereby enhancing soil fertility (Yuan et al., 2023; Vitali et al., 2024). As the most efficient method for straw utilization, conservation tillage prevents pollution from open burning and releases nitrogen, phosphorus, potassium, and trace elements during residue decomposition. This process improves soil structure, water-nutrient availability, and carbon sequestration capacity, ultimately benefiting root growth and yield (Jin et al., 2017). Research on conservation tillage in maize-soybean rotations is thus essential to address straw management challenges, improve soil quality, and ensure sustainable agriculture in Northeast China's mollisol region.

No-tillage with stubble mulching (NTMS), an advanced mechanized tillage-sowing system, builds upon conventional no-tillage sowing. While plowing loosens soil, modifies physical structure, and reduces soil bulk density (SBD) and compaction to enhance crop growth (Lampurlanés and Cantero‐Martínez, 2003; Suslov and Biochemistry, 2020), it may retard straw decomposition and reduce soil water retention relative to no-tillage during maize cultivation (Li et al., 2021). Rotary tillage promotes uniform straw distribution but often induces plow pan formation below the tillage layer, restricting root penetration and nutrient uptake (Thorup-Kristensen et al., 2020). Unlike conventional tillage that disrupts aggregates, no-tillage maintains structural integrity through minimal disturbance, concurrently elevating SWC for sustainable agroecosystem management (Jabro et al., 2016; Yang et al., 2018; Souza et al., 2021). However, prolonged no-tillage may reduce wheat yields, increase BD, and decrease soil fertility through shallow tillage layers (Ernst et al., 2016). Surface residue retention in NTMS enhances hydraulic properties and agronomic productivity relative to no-tillage alone (Haruna et al., 2018).

Growing soil conservation awareness has spurred adoption of no-tillage and NTMS. Both technologies employ direct seeding on untilled soil, minimizing structural disruption from traditional tillage (Etana et al., 2020). Specialized NTMS machinery reduces open-field straw burning (improving air quality), enhances moisture retention via straw cover (conserving water), and lowers production costs by eliminating tillage steps (Ma et al., 2025). NTMS mitigates soil nitrogen loss, preserves fertility, and maintains yields while reducing fertilizer inputs, thereby countering degradation (Zhang et al., 2025a) . In semi-arid Australia, NTMS outperforms no-tillage in boosting SWC during wheat fallow, though outcomes vary with soil composition (O'leary and Connor, 1997) . In karst mountainous regions, no-tillage with hairy vetch cover elevates rhizosphere/non-rhizosphere enzyme activities, increases available nitrogen, optimizes nitrogen utilization, and raises silage maize yields while lowering costs (Li et al., 2024b). High-stubble NTMS improves nutrient availability, reshapes microbial communities, enhances diversity, and elevates soil quality and crop yields (Zhang et al., 2025b) .





Nevertheless, existing studies overlook agricultural machinery effects on soil penetration resistance (SPR)—a critical factor undermining conservation efforts in Northeast China's mollisol region. SPR is a key indicator of soil compaction, directly affecting structure and productivity (Håkansson et al., 2000). Compaction reduces porosity by compressing soil particles, thereby increasing soil bulk density (SBD) (Kumar et al., 2018). It also impairs pore connectivity, restricting root development, nutrient-water uptake, seedling emergence, and plant height, ultimately diminishing crop yields (Place et al., 2008; Ramos et

al., 2022). Significant SPR effects on corn height and yield have been documented (Rasche Alvarez et al., 2020). Tillage practices effectively alleviate SPR, countering compaction impacts; winter wheat pre-sowing plowing notably reduces SPR (Ren et al., 2018). Thus, systematic evaluation of mechanized tillage-sowing effects on SPR is vital for optimizing conservation strategies and mitigating degradation.

This study aims to: i) Conduct a comparative experiment under maize-soybean rotation in Northeast China's mollisol region, evaluating four mechanized practices: plow tillage with precise sowing (PTS), rotary tillage with precise sowing (RTS), no-tillage sowing (NTS), and no-tillage with stubble mulching and sowing (NTMS); ii) Employ SPR as a primary indicator to assess tillage effects on compaction, while examining impacts on SWC and SBD. Results will provide technical insights for optimizing tillage systems, mitigating degradation, and advancing sustainable agriculture.

## 80    2 Materials and Methods

### 2.1 Experimental sites

The experiment was conducted at the Xiangyang Experimental Base of Northeast Agricultural University (44°04'N, 125°42'E) in Xiangfang District, Harbin, China. The site features typical black soil with a silt loam texture, composed of 5.3% sand, 68.5% silt, cand 28.9% clay. Located within Heilongjiang Province's primary thermal zone, the base experiences a temperate

continental monsoon climate. Between 2017 and 2021, the mean annual temperature was 5.4°C with an average frost-free period of 145 days. The trial period spanned October 2021 to October 2022, with key parameters measured during four critical growth stages in 2022. Average daily rainfall and weekly temperature from January to October 2022 are presented in Figure 1.





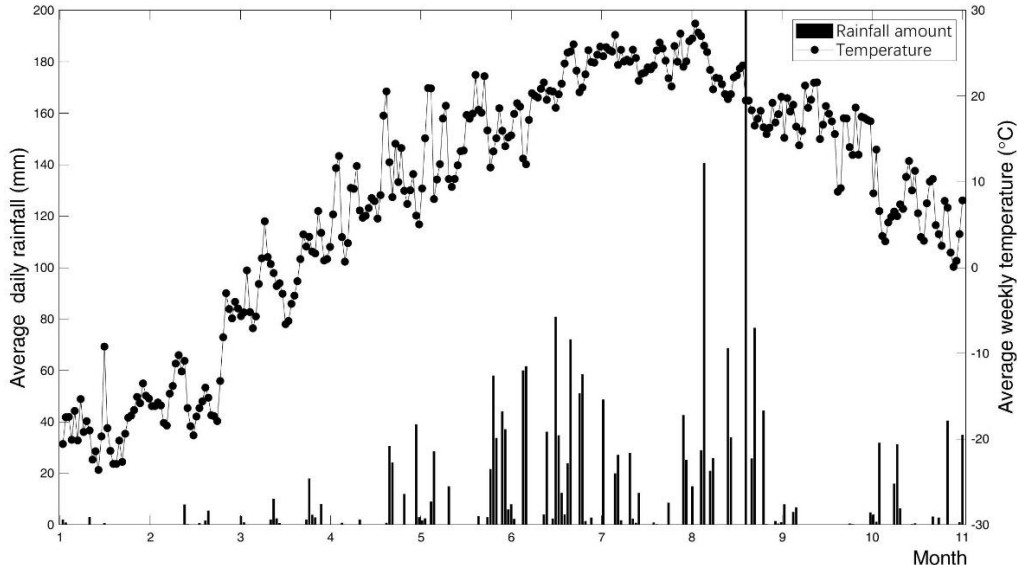

Figure 1: Daily rainfall and weekly average temperature from January to November in 2022.

## 2.2 Experiment design

The experiment comprised four mechanized tillage-sowing practices (PTS, RTS, NTS, NTMS) implemented within a maize-soybean rotation system aligned with actual field practices in Heilongjiang Province. Specific operational details are presented in Table 1.

| Practices | Detail mechanized operations |
|---|---|
| plow tillage and precision sowing (PTS) | During autumn 2021, moldboard plowing turned over soil and buried crushed maize straw to 25-30 cm depth, followed by two rotary tillage passes, ridging, and compaction. Soybeans were sown using a precision seeder in spring 2022. |
| rotary tillage and precision sowing (RTS) | After subsoiling, rotary tillage crushed and mixed maize straw with soil in the 0-15 cm layer during autumn 2021, followed by ridging and compaction. Soybeans were precision-sown in spring 2022. |
| no-tillage sowing (NTS) | Soybeans were direct-sown with a no-tillage seeder in spring 2022 without prior tillage operations. |



| no-tillage with stubble mulching and sowing (NTMS) | Maize straw was retained post-harvest in autumn 2021 without tillage. In spring 2022, soybeans were direct-sown using a 2BMFJ precision seeder with maize stubble maintained on the soil surface (Hou et al., 2022). |
|---|---|

**Table 1.** Text plan.

A randomized complete block design with three replicates was implemented to distribute plots for different tillage-sowing practices. Each plot measured 8 m × 3.9 m (six ridges), with a central testing area of 2.6 m width (four ridges) and 0.65 m protective borders (one ridge per side). A 6-m buffer zone between vertically adjacent plots accommodated machinery turning.

The total experimental area spanned 48 m × 15.6 m, covering 748.8 m². Within each plot, sampling points were established at four soil depths (5, 15, 25, and 35 cm) using cutting ring methods. The experimental layout is illustrated in Figure 2.

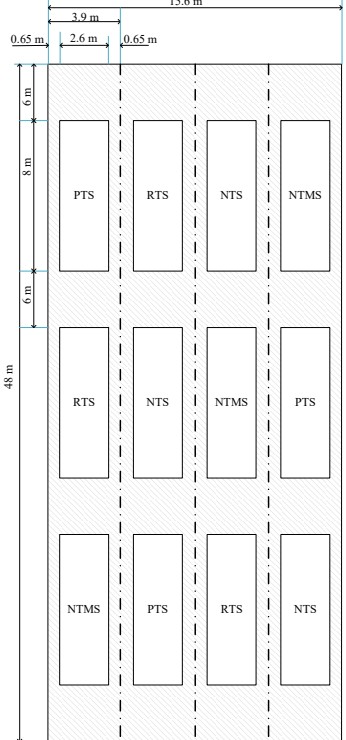

**Figure 2: Layout of the experimental plots.**

Soil samples were collected during four soybean growth stages: the post-sowing period (May), emergence (June), podding

(August), and maturity (September). Sampling occurred on precipitation-free days following ≥7 consecutive rain-free days.





## 2.3 Methods for indicator testing and data-analysis

### 2.3.1 Determination Method of Soil Water Content

Soil water content (SWC) refers to the water fraction or percentage in a given mass of soil. The aluminium dish drying method was used to determine soil water content in this experiment. The soil water content which can be calculated as shown in Eq.

(1) (NY/T52-1987).

$$SWC = \frac{(m_1 - m_2)}{(m_2 - m_0)} \times 100\%$$   (1)

where $m_0$ is mass of the aluminium container $[g]$ , $m_1$ is mass of fresh soil and aluminium container before drying $[g]$ , $m_2$ is mass of dried soil and aluminium container after drying $[g]$ .

### 115   2.3.2 Determination Method of Soil Bulk Density

Soil Bulk Density (SBD) is defined as the ratio of the mass of dry soil solids to the total volume of the soil sample. This value is also referred to as soil density. The formula for calculating SBD as shown in Eq. (2) (NY/T1121.4-2006).

$$SBD = \frac{100 \times (m_1 - m_0)}{V (100 + SWC)}$$   (2)

where $m_0$ is mass of the cutting ring $[g]$ , $m_1$ is mass of fresh soil and cutting ring before drying $[g]$ ,

$V$ is volume of the cutting ring, is 100 $[cm^3]$ .

### 2.3.3 Determination Method of Soil Penetration Resistance

Soil penetration resistance (SPR), a key indicator of soil compaction, was measured using an Eijkelkamp PV 6.08 penetrometer (Netherlands) featuring a 60° conical tip with 1 cm² base area and 100-cm rod. Measurements extended to 80 cm depth. Within

each plot, a random transect was selected for SPR assessment with seven measurement points spaced at 11-cm intervals. During operation, the instrument was maintained horizontally and inserted vertically at 5 cm/s.

## 3 Results

### 3.1 Soil Water Content (SWC)

The significance analysis of SWC under different mechanized tillage and sowing practices across four testing periods were shown in Figure 3.





### 3.1.1 The effect of different mechanized tillage and sowing practices on SWC at same depth

(1) Post-sowing stage

Aat the depth of 5 cm, there was no significant difference in SWC among NTMS, PTS and RTS. But the SWC of NTMS

was significantly higher than that of NTS by 10% (p<0.05). At the depth of 15 cm and 35 cm, there were no significant differences in SWC among practices. However, at 25 cm, the SWC of NTMS practice was significantly higher than that of the NTS, PTS, and RTS by 8.8%, 10.7%, and 13.5%, respectively (p<0.05).

(2) Emergence stage

There were no significant differences were found in SWC among practices at the depth of 5 cm. At the depth of 15 cm, no significant difference in SWC was observed among NTMS and NTS, RTS. But the SWC of NTMS practice was significantly higher than that of PTS practice by 23.5% (p<0.05). At the depth of 25 cm, the SWC of NTMS practice was significantly higher than that of NTS, RTS, and PTS by 10.1%, 12.1%, and 18.4%, respectively (p<0.05). At the depth of 35 cm, no significant differences of SWC were observed among NTMS and RTS, NTS practices, but the SWC of NTMS practice was

significantly higher than that of PTS by 7.2% (p<0.05).

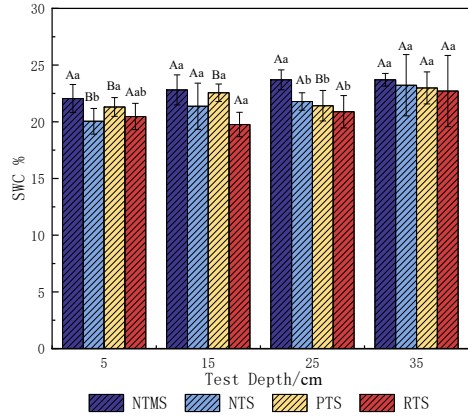

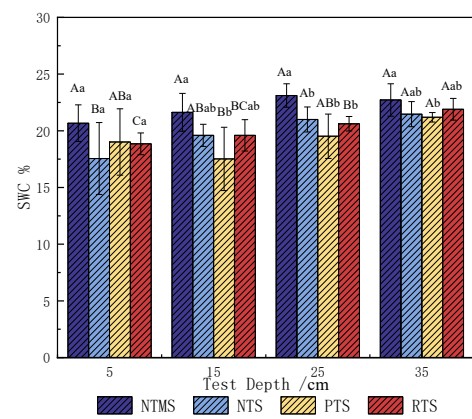

a) The post-sowing stage          b) The emergence stage





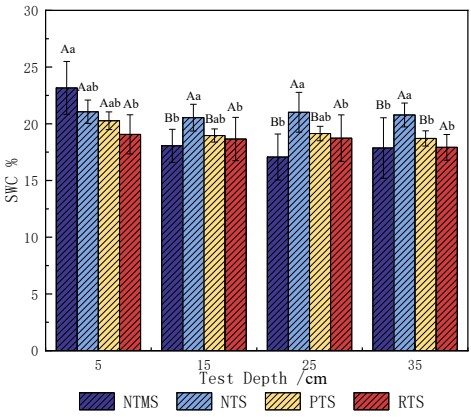
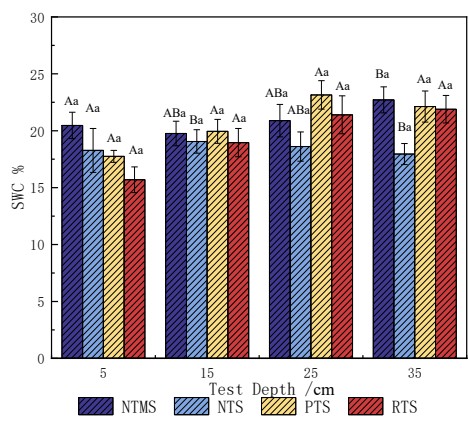

c) The podding stage                    d) The maturity stage


**Figure 3: Significance analysis of SWC among different depths and different mechanized tillage and sowing practices. Note: Uppercase letters denote significant differences among various soil depths within the same mechanized tillage and sowing practice (p<0.05); lowercase letters signify significant differences between distinct mechanized tillage and sowing practices at the same soil depth (p<0.05). The same below.**

(3) Podding stage

At the depth of 5 cm, the SWC of NTMS practice was significantly higher than that of RTS by 21.5% (p<0.05). At the depth of 15 cm, the SWC of NTS practice was significantly higher than that of NTMS and RTS by 13.7% and 10.1%, respectively (p<0.05). At the depth of 25 cm, no significant differences of SWC were found among the mechanized tillage and sowing practices. At the depth of 35 cm, the SWC of NTS practice was significantly higher than that of PTS, RTS, and NTMS by

11.1%, 15.9%, and 16.3%, respectively (p<0.05).

(4) Maturity stage

Ther were no significant differences of SWC were found among the mechanized tillage and sowing practices.

**3.1.2 The effect of same mechanized tillage and sowing practices on SWC at different depths**

The significance analysis of SWC under different mechanized tillage and sowing practices across four testing periods is shown in Figure 3.

(1) NTMS

During the post-sowing stage of soybean, there were no significant differences in SWC among the different soil depths. At the

emergence stage, there were also no significant differences in SWC among various soil depths. At the podding stage, the SWC at 5 cm depth was the highest, significantly higher than at 15 cm, 25 cm, and 35 cm depths by 28.3%, 35.7%, and 29.7%,





respectively (p < 0.05). At the maturity stage, the SWC at 35 cm depth was the highest, significantly higher than at 5 cm depth by 11% (p < 0.05).

(2) NTS

During the post-sowing stage of soybean, SWC increased with increasing soil depth, with the SWC at 5 cm depth being significantly lower than at 15 cm, 25 cm, and 35 cm depths by 6.2%, 8.0%, and 13.7%, respectively (p < 0.05). At the emergence stage, the SWC at 5 cm depth was significantly lower than at 25 cm and 35 cm depths by 16.4% and 18.2%, respectively (p < 0.05). During the podding stage, there were no significant differences in SWC among different soil depths.

At the maturity stage, the SWC at 5 cm depth was significantly lower than that at 15 cm depth by 4.1% (p < 0.05) and significantly higher than that at 35 cm depth by 2.3% (p < 0.05).

(3) PTS

During the post-sowing stage of soybean, the SWC at 35 cm depth was the highest, significantly higher than 5 cm, 15 cm, and

25 cm depths by 7.9%, 1.9%, and 7.3%, respectively (p < 0.05). At the emergence stage, the SWC at 35 cm depth was also the highest, significantly higher than at 15 cm depth by 21% (p < 0.05). At the podding stage, the SWC at 5 cm depth was the highest, significantly higher than at 15 cm, 25 cm, and 35 cm depths by 6.9%, 5.9%, and 8.4%, respectively (p < 0.05). At the maturity age, there were no significant differences in SWC among the various depths.

(4) RTS

During the post-sowing stage of soybean, no significant differences in SWC were observed among the various depths. At the emergence stage, the SWC at 35 cm depth was the highest, significantly higher than at 5 cm, 15 cm, and 25 cm depths by 16.2%, 11.8%, and 6.2%, respectively (p < 0.05). The SWC at 25 cm depth was also significantly higher than at 5 cm depth by 9.4% (p < 0.05). During the podding and the maturity stages, no significant differences in SWC were noted among the

different depths.

**3.2 Soil Bulk Density**

The significance analysis of SBD under different mechanized tillage and sowing practices across four testing periods were shown in Figure 4.



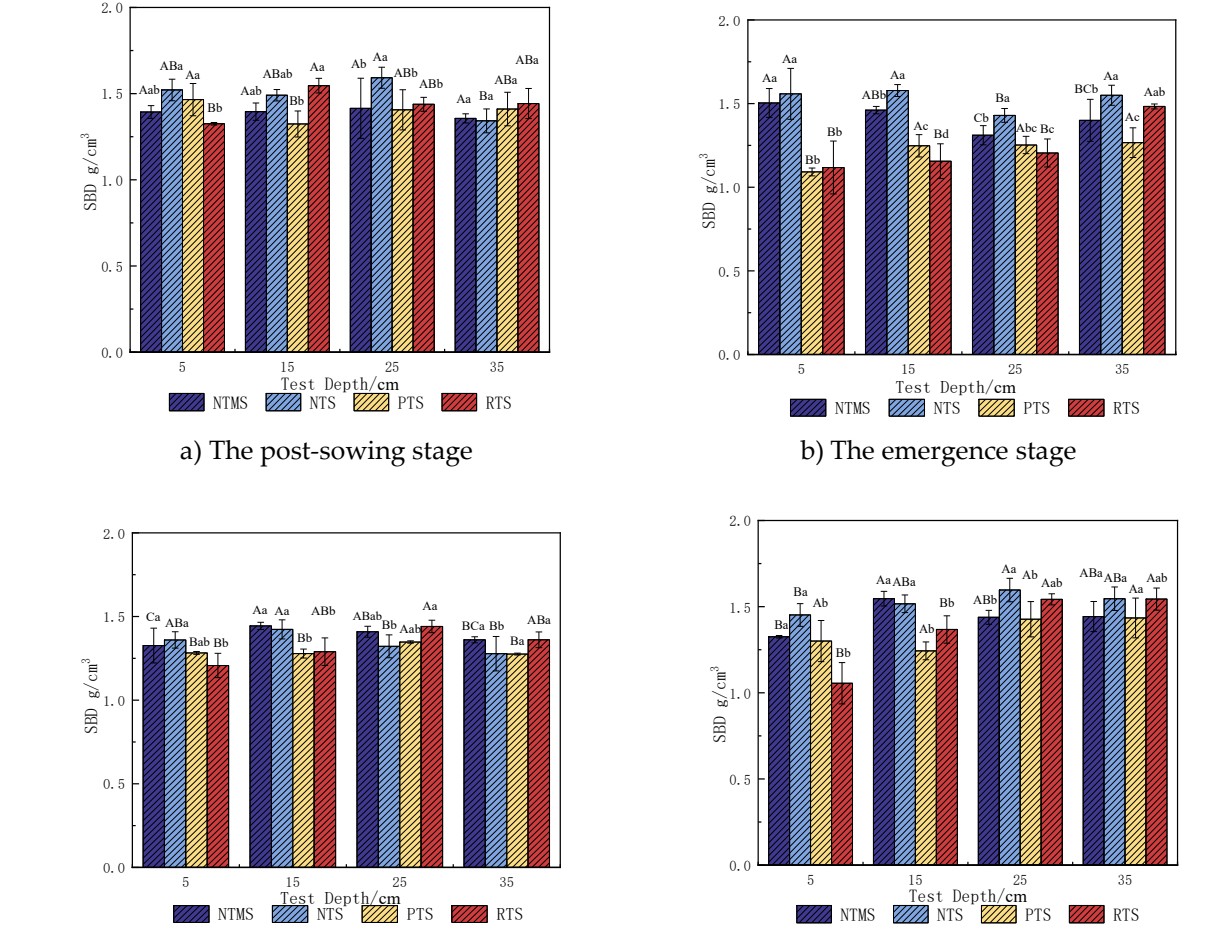

a) The post-sowing stage

b) The emergence stage

c) The podding stage

d) The maturity stage

**Figure 4: Significance analysis of soil bulk density at different depths for different mechanized tillage and sowing practices**

**3.2.1 The effect of different mechanized tillage and sowing practices on SBD at the same depth**

(1) Post-sowing stage

At the depth of 5 cm, the SBD of NTS practice showed no significant difference compared to that of NTMS and PTS. The SBD of NTS was significantly higher than that of RTS by 14.8% (p < 0.05). At the depth of 15 cm, the SBD of RTS practice exhibited no significant differences from that of NTS and NTMS practices. The SBD of RTS was significantly higher than that of PTS by 16.8% (p < 0.05). At the depth of 25 cm, the SBD of NTS practice was significantly higher than that of NTMS, PTS, and RTS by 12.6%, 13.3%, and 10.7%, respectively (p < 0.05). No significant differences in SBD were noted among the mechanized tillage practices at the depth of 35 cm.

(2) Emergence stage



At the depth of 5 cm, the SBD of NTS practice was significantly higher than that of PTS and RTS by 42.7% and 39.4%, respectively ($p < 0.05$), while the SBD of NTMS practice was significantly higher than that of PTS and RTS by 37.8% and 34.5%, respectively ($p < 0.05$). At the depth of 15 cm, the SBD of NTS practice was significantly higher than that of NTMS, PTS, and RTS by 8.0%, 26.5%, and 36.5%, respectively ($p < 0.05$). The SBD of NTMS practice was also significantly higher than that of PTS and RTS by 17.1% and 26.4%, respectively ($p < 0.05$). At the depth of 25 cm, the SBD of NTS practice was significantly higher than that of NTMS, PTS, and RTS practices by 9.0%, 14.1%, and 18.6%, respectively ($p < 0.05$). The SBD of NTMS practice was significantly higher than that of PTS and RTS practices by 4.7% and 8.8% ($p < 0.05$). At the depth of 35 cm, the SBD of NTS practice was significantly higher than that of NTMS and PTS practices by 10.7% and 22.4% ($p < 0.05$). The SBD of NTMS practice was significantly higher than that of PTS practice by 10.5% and 16.5% ($p < 0.05$).

(3) Podding stage

At the depth of 5 cm, the SBD of RTS practice was significantly lower than that of NTS and NTMS by 11.2% and 9.0% ($p < 0.05$). At the depth of 15 cm, the SBD of NTMS practice was significantly higher than that of PTS and RTS practices by 13% and 12%, respectively ($p < 0.05$). The SBD of NTS practice was also significantly higher than that of PTS and RTS by 11.3% and 10.4%, respectively ($p < 0.05$). At the depth of 25 cm, the SBD of RTS practice was significantly higher than that of NTS by 9% ($p < 0.05$). At the depth of 35 cm, the SBD of NTMS practice was significantly higher than that of NTS and PTS by 6.5% ($p < 0.05$). The SBD of RTS practice was also significantly higher than that of NTS and PTS by 6.5% and 6.7%, respectively ($p < 0.05$).

(4) Maturity stage

At the depth of 5 cm, the SBD of NTMS practice was significantly higher than that of PTS and RTS by 2.3% and 25.5%, respectively ($p < 0.05$). At the depth of 15 cm, the SBD of NTS practice was significantly higher than that of PTS and RTS by 22.6% and 10.9%, respectively ($p < 0.05$), and the SBD of NTMS practice was significantly higher than that of PTS and RTS by 25.0% and 13.1%, respectively ($p < 0.05$). At the depth of 25 cm, the SBD of NTS practice was significantly higher than that of NTMS and PTS by 11.1% and 11.9%, respectively ($p < 0.05$). At the depth of 35 cm, no significant differences in SBD were observed among all practices.

**3.2.2. The effect of same mechanized tillage and sowing practices on SBD at different depths**

The significance analysis of SBD under different mechanized tillage and sowing practices among four testing periods is shown in Figure 4.

(1) NTMS

During the post-sowing stage of soybean, no significant differences in SBD were observed among different soil depths. At the emergence stage, the SBD at 25 cm depth was significantly lower than at 5 cm and 15 cm depths by 12.8% and 10.3%





respectively (p < 0.05). The SBD at 5 cm depth was significantly higher than at 35 cm depth by 7.4% (p < 0.05). During the podding stage, the SBD at 5 cm depth was significantly lower than at 15 cm and 25 cm depths by 8.2% and 5.9% respectively (p < 0.05). The SBD at 15 cm depth was significantly higher than at 35 cm depth by 6.1% (p < 0.05). At the maturity stage,

the SBD at 15 cm depth was significantly higher than at 5 cm depth by 16.5% (p < 0.05). No significant differences in SBD were observed in other practices.

(2) NTS

During the post-sowing stage of soybean, the SBD at 25 cm depth was significantly higher than at 35 cm depth by 18.7% (p <

0.05). No significant differences in SBD were observed in other practices. At the emergence stage, the SBD at 25 cm depth was significantly lower than at 5 cm, 15 cm, and 35 cm depths by 8.3%, 9.4%, and 7.8% respectively (p < 0.05). No significant differences in SBD were observed in other practices. During the podding stage, the SBD at 15 cm depth was significantly higher than at 25 cm and 35 cm depths by 7.7% and 11.4% respectively (p < 0.05). No significant differences in SBD were observed in other practices. At the maturity stage, the SBD at 25 cm depth was significantly higher than that at 5 cm depth by

10.3% (p < 0.05). No significant differences in SBD were observed in other practices.

(3) PTS

During the post-sowing stage of soybean, the SBD at 5 cm depth was significantly higher than at 15 cm depth by 10.7% (p < 0.05). No significant differences in SBD were observed in other practices. At the emergence stage, the SBD at 5 cm depth was

significantly lower than that at 15 cm, 25 cm, and 35 cm depths by 12.5%, 12.9%, and 13.8% respectively (p < 0.05). No significant differences in SBD were observed in other practices. During the podding stage, the SBD at 25 cm depth was significantly higher than that at 5 cm, 15 cm, and 35 cm depths by 5.1%, 5.4%, and 5.6% respectively (p < 0.05). No significant differences in SBD were observed in other practices. At the maturity stage, no significant differences were observed among the soil depths.


(4) RTS

During the post-sowing stage of soybean, the SBD at 15 cm depth was significantly higher than at 5 cm depth by 16.7% (p < 0.05). No significant differences in SBD were observed in other practices. At the emergence stage, the SBD at 35 cm depth was significantly higher than at 5 cm, 15 cm, and 25 cm depths by 32.7%, 28.3%, and 23.1% respectively (p < 0.05). The SBD

at 25 cm depth was also significantly higher than at 5 cm depth by 9.4% (p < 0.05). During the podding stage, the SBD at 25 cm depth was significantly higher than at 5 cm depth by 19.4% (p < 0.05). No significant differences in SBD were observed in other practices. At the maturity stage, the SBD at 35 cm depth was significantly higher than at 5 cm and 15 cm depths by 45.3% and 13.1% respectively (p < 0.05). No significant differences in SBD were observed in other practices.





### 3.3. Soil Penetration Resistance

The variations in SPR under different mechanized tillage and sowing methods at various growth stages are illustrated in Figure 5. It can be observed that the SPR of the four mechanized tillage and sowing practices maintained a similar trend throughout the soybean growth cycle, with the SPR maintaining a relatively high level during the pod-ding stage of soybean.

### 3.3.1 Variation of SPR in cross sections during different stages

(1) Post-sowing stage

As depicted in Figure 5 A, it was showed that SPR is maximized under NTS practice. The SPR under the NTS practice (b) ranges from 0.5 to 2.2 MPa at depths of 0 to 30 cm, occurring at measurement points from -11 cm to 11 cm. The maximum SPR value exceeded that of other practices by 10% to 57%. Across all practices, the SPR increases with depth and shows an upward trend from the central measurement point toward both sides. For NTS practice, peaking at depths of 60 to 80 cm, at measurement points from -11 cm to 0 cm, where it reaches a maximum value of 2.8 MPa. For RTS practice, the area with a

higher SPR occurs at the depth of 60-80 cm, specifically at measurement points from -33 cm to -22 cm and 11 cm to 22 cm, where the value ranges from 1.7 to 2.6 MPa. For PTS practice, the area with a higher SPR occurs at the depth of 60-80 cm, specifically at measurement points from -33 cm to -22 cm and 11 cm to 33 cm, where the value ranges from 1.9 to 2.2 MPa. For NTMS practice, the areas of higher SPR are mainly concentrated on the right side of the centre measurement point, specifically at measurement points from 11 cm to 33 cm, where the values range from 1.5 to 1.7 MPa.


(2) Emergence stage

Figure 5 B was showed that the SPR of NTS practice was higher than other practices. The SPR under NTS practice (b) ranges from 0.5 to 2.2 MPa at depths of 0 to 30 cm which the maximum SPR value exceeded that of other practices by 37.5% to 57.1%. Across all practices, the SPR increases with depth and shows an upward trend from the central measurement point

toward both sides. For NTS practice, the highest SPR occurs at the depth of 60-80 cm, specifically at measurement points from 22 cm to 33 cm and -11 cm, where the value reaches 2.9 MPa. The highest SPR value of NTS practice exceeding the maximum values of other practices by 45% to 70.6%. For RTS practice, the area with a higher SPR occurs at the depth of 40-70 cm and 60-80 cm, specifically at measurement points from -33 cm to -11 cm and 11 cm to 33 cm, where the value ranges from 1.7 to 2.0 MPa. For PTS practice, the area with a higher SPR occurs at the depth of 30-80 cm, specifically at measurement points

from 11 cm to 33 cm, where the value ranges from 1.7 to 1.9 MPa. For NTMS practice, the area with a higher SPR occurs at the right side of the center measurement point between 11 and 22 cm, at the depths of 20-30 cm where the value ranges from 1.7 to 1.9 MPa.

(3) Podding stage





Figure 5 C was showed that the SPR of NTMS practice was higher than other practices. The SPR for NTMS (a) ranges from 0.6 to 4.9 MPa at depths of 0 to 40 cm, demonstrating a trend of initially increasing. This trend was also observed in the PTS and RTS practices. The highest SPR occurs at the depth of 10-40 cm, specifically at the measurement point at -22 cm, where the value reaches 4.9 MPa. The highest SPR value of NTMS practice exceeding the maximum values of other practices by 8.9% to 53.1%. For PTS practice, the higher values mainly concentrated at depths of 20 to 70 cm, specifically at measurement

points from -33 cm to -22 cm and 22 cm. For RTS practice, the area with a higher SPR occurs at the depth of 20-40 cm, specifically at measurement points from -33 cm to -11 cm, where the value reaches 4.5 MPa. For NTS practice, the SPR increases with depth. The area with a higher SPR occurs at the depth of 50-80 cm, specifically at the measurement point at 22 cm, where the value reaches to 3.2 MPa.

(4) Maturity stage

Figure 5 D was showed that the SPR of NTS practice was higher than other practices. The highest SPR occurs at the depth of 0-10 cm, specifically at measurement points from 11 cm to 33 cm, where the value reaches 3.9 MPa. The highest SPR value of NTS practice exceeding the maximum values of other practices by 20% to 28.6%. Across NTS, PTS and NTMS practices, the SPR increases with depth. Across all practices was shows an upward trend from the central measurement point toward both

sides. RTS practice shows a trend of initially increasing and then decreasing SPR. The highest SPR occurs at the depth of 10-30 cm, specifically at the measurement point at 22 cm, where the value reaches 3.0 MPa. For PTS practice, the highest SPR occurs at the depth of 10-20 cm, specifically at measurement points from 11 cm to 33 cm, where the value reaches 3.0 MPa. For NTMS practice, the highest SPR occurs at the depth of 0-20 cm, specifically at measurement points from -33 cm to 11 cm, where the value reaches 2.8 MPa.






**Figure 5:Soil penetration resistance at each soil measurement point at different periods under different mechanized tillage and sowing practices.**



**335  3.3.2 The effect of the same mechanized tillage and sowing practices at the same depth on SPR in different stages**

Under the same mechanized tillage and sowing practices, the changes in SPR during testing periods were illustrated in Figure 6.

SPR was found to be the highest during the post-sowing stage among four practices, particularly concentrated in the soil depth

of over 40 cm. SPR at the maturity stage under four practices was greater than that observed during the post-sowing stage, with an increase ranging from 8.5% to 46.2%.

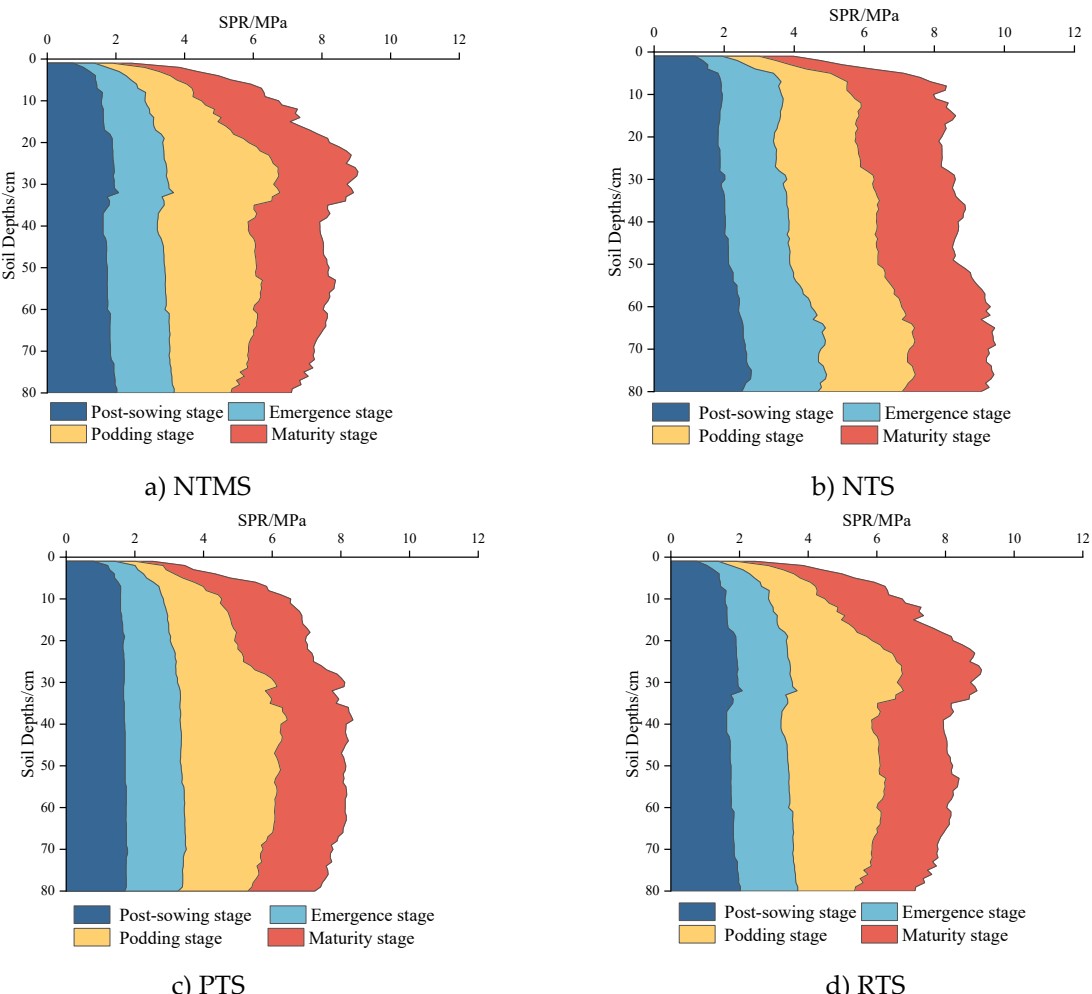


**Figure 6:  Soil penetration resistance under same mechanized tillage and sowing practices under different stages.**

Under the NTMS practice, SPR reached its maximum during the podding stage, being 26.7%–97.0% higher than in other stages, while it was primarily concentrated at soil depths exceeding 30 cm. During the emergence stage under NTMS practice, SPR was at its minimum, lower than other stages by 6.0% to 49.5%. For the NTS practice, the maximum SPR was observed





during the podding stage, higher than other stages by 17.5% to 100.2%. The emergence stage under NTS practice also exhibited the lowest SPR, lower than other stages by 15.0% to 50.1%, with SPR distribution being relatively uniform among stages. In the PTS practice, SPR was highest during the podding stage, higher than other stages by 10.1% to 50.7%, concentrated at soil depths exceeding 40 cm. The emergence stage under PTS practice was exhibited the lowest SPR, lower than other stages by 9.2% to 33.6%. In the RTS practice, the seedling stage showed the lowest SPR, lower than other stages by 6.0% to 49.5%, while the podding stage displayed the highest SPR, higher than other stages by 13.5% to 55.8%, concentrated at soil depths exceeding 35 cm.

## 4. Discussion

### 4.1. Soil Water Content

During the post-sowing stage of soybeans, NTMS exhibited significantly higher SWC than NTS at 5 cm and 25 cm depths (post-sowing), and at 25 cm and 35 cm depths (emergence). This results from straw mulch reducing surface evaporation and runoff. Mechanized no-till with straw mulching enhances SWC in NTMS versus NTS by suppressing moisture evaporation (Hou and Li, 2018). Although NTS retains subsurface residues, the absence of surface coverage limits SWC improvement during early growth stages.

During the podding stage, at 15 cm depth, NTS showed significantly higher SWC than NTMS and RTS, while at 35 cm depth it exceeded all other practices. Surface residues in NTS slow runoff, reduce evapotranspiration, and enhance water infiltration/storage (Martínez et al., 2008; Alhameid et al., 2019). Since the podding stage occurs during the rainy season, this may explain why the SWC in deeper soil layers under the NTS practice is significantly higher than in other practices at this stage.

Compared to the SWC of PTS and RTS practices, the NTMS practice had significantly higher SWC at 25 cm depth during the post-sowing stage. At the emergence stage, the SWC of NTMS practice was significantly higher at 15 cm and 25 cm depth than that of PTS practice, and at 25 cm depth, it was also significantly higher than that of RTS practice. During the podding stage, the SWC of NTMS practice at 5 cm depth was significantly higher than that of RTS practice. This indicates that NTMS practice, compared to PTS and RTS practices, can increase SWC, with a more significant improvement observed at soil depths of 5 cm and 25 cm. The consistency between this finding and existing literature underscores the robustness of NTMS practice can enhances SWC more than PTS and RTS practice during some stages (Chen et al., 2024).

Conservation tillage is beneficial for increasing soil water content and water use efficiency in farmland (Choudhury et al., 2014). Both PTS and RTS practices also contribute to improving SWC. The reason lies in the straw buried underground in the PTS practice, whose loose structure increases soil permeability and has a higher water absorption capacity, thereby enhancing





the soil's ability to conduct and retain water (Zhao et al., 2022). In the RTS practice, sub-soiling reduces the evaporation losses of soil moisture caused by traditional moldboard plowing, improves soil infiltration capacity, and increases the saturated hydraulic conductivity, thus raising SWC (Li et al., 2024a). This indicates that both PTS and RTS practices are conducive to 385 enhancing the water retention capacity of deeper soil layers, which also explains the lack of significant difference between the PTS and RTS practices in this experiment.

## 4.2. Soil Bulk Density

During the post-sowing and maturity stages of soybeans, the SBD at 25 cm depth of NTS practice was significantly higher than that of NTMS practice. This indicates that, at 25 cm depth, NTMS practice is more effective than NTS in reducing SBD. 390 During the emergence stage, the SBD at 15 cm depth of NTS practice was significantly higher than of NTMS practice. The difference between NTS and NTMS can be attributed to the straw cover in the NTMS practice, which affects soil structure and environment, increases soil organic matter, and enhances soil infiltration capacity (Chen Yuzhang et al., 2019).

During the post-sowing stage, the SBD at 5 cm depth of NTS practice was significantly higher than of RTS practice; the SBD 395 at 25 cm depth of NTS practice was significantly higher than in the PTS and RTS practices. During the emergence stage, the SBD at 5 cm and 15 cm depths of NTS practice was significantly higher than in the PTS and RTS practices, and during both the podding and maturity stages, the SBD of NTS practice was significantly higher than of PTS and RTS practices at all depths. This suggests that certain tillage practices can effectively reduce SBD. Related studies have shown that plowing and subsoiling can significantly reduce SBD and shear strength, increase the thickness of the plow layer, and improve soil porosity (Li et al., 400 2021), which may explain why the PTS and RTS practices maintained lower average SBD compared to NTS.

During the emergence stage the SBD at 5 cm, 15 cm, and 25 cm depths of NTMS practice was significantly higher than in the PTS and RTS practices. During the podding stage, SBD at 5 cm and 35 cm depths of NTMS practice was significantly higher than that of PTS and RTS. And during the maturity stage, SBD of NTMS practice was significantly higher than of PTS and 405 RTS practices. This suggests that, compared to the more disruptive PTS and RTS practices, the NTMS practice, which causes less disturbance to the soil structure, is less effective in reducing SBD. Related studies have shown that no-tillage causes minimal disturbance to the soil structure and can significantly increase SBD (Li et al., 2021), which is consistent with the results of this study.

Due to the disruption of soil structure in the PTS and RTS practices, the SBD between the two practices remained mostly at an insignificant level throughout different stages. Related research indicates that plowing before sowing can significantly reduce SBD and increase soil porosity (Abidela Hussein et al., 2019). This can also explain why the SBD at 15-25 cm depths of PTS practice during the soybean growth stage was significantly lower than that of NTS practice. And the SBD at 25 cm depth of PTS practice was significantly lower than that of NTMS practice. The rotary tillage in the RTS practice, which is





shallower than that in the PTS practice, is less effective in reducing SBD. That may explain why at 35 cm depth during the podding stage, the SBD in the RTS practice was significantly higher than in the PTS practice.

### 4.3. Soil Penetration Resistance

Compared to other practices, SPR after NTS practice was relatively high, which is consistent with some research findings (De Oliveira et al., 2022). The higher SPR in the NTS practice is due to minimal disturbance to the soil structure in no-tillage

practices, which significantly increases soil bulk density, leading to greater soil compaction. Related studies have shown that no-tillage or reduced tillage can promote biological activity and improve soil structural quality (De Oliveira et al., 2022). However, after long periods of no-till or reduced tillage, the soil's physical conditions become similar to tthe physical conditions under conventional tillage practices (Barbosa et al., 2019; Da Luz et al., 2022).

The four mechanized tillage and sowing practices maintained a similar trend throughout the soybean growth cycle. This could be because two to three years of continuous different tillage practices did not result in significant differences in soil physical properties and crop yield (De Oliveira et al., 2022). However, the NTMS practice exhibited lower SPR before the soybean emergence stage compared to the other practices. This is likely because the straw cover influenced soil structure and the soil environment. Research by Castioni also indicated that maintaining crop residues on the soil surface helps protect soil physical

quality (Castioni et al., 2019), reducing wheel pressure on the soil, which in turn decreases soil bulk density and SPR (Reichert et al., 2016). Over time, SPR in the NTMS practice gradually increased and was higher than in the other practices during the podding stage. The changes in SPR between the PTS and RTS practices were similar, but the SPR of PTS practice was lower than that of the RTS practice. This can be attributed to the rotary tillage in the RTS practice, which increases SBD, thereby increasing SPR. Related studies have also shown that deep tillage loosens deeper soil layers and reduces soil compaction

compared to no-till and rotary tillage (Lou et al., 2021). In the RTS practice, subsoiling is also performed after rotary tillage, which may explain why the changes in SPR were similar for the PTS and RTS practices in deeper soil layers, e.g., below 40 cm.

### 5. Conclusions

The results demonstrate that tillage-sowing practices significantly influence soil physical properties. Compared to other

practice, NTMS practice increased SWC, whereas PTS and RTS exhibited no significant differences. NTS practice was less effective in reducing SBD. During soybean post-sowing and emergence stages, NTMS showed significantly lower SBD at 15 cm and 25 cm depths than NTS. At 25 cm depth, PTS and RTS displayed non-significant SBD variations. NTS resulted in higher SPR than other practices. Before soybean emergence, NTMS consistently yielded the lowest SPR. Across all mechanized tillage-sowing practices, SPR trends remained similar throughout the soybean growth cycle, peaking during the

podding stage.



The results demonstrated that the NTMS practice is feasible for improving soil structure, protect black soil resources and addressing the issue of excess maize straw in the maize-soybean rotation system in the mollisol region of Northeast China.

**Author contributions.**

Conceptualization, J.Q.; methodology, J.Q. and D.W.; data curation, D.W., J.S., H.S., L.W. and B.W.; formal analysis, writing—original draft preparation, and visualization, D.W.; Writing—review and editing, J.Q. and D.W.; funding acquisition, J.Q. and H.C. All authors have read and agreed to the published version of the manuscript.

**Competing interests.**

The contact author has declared that none of the authors has any competing interests.

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
