# Peer review of "No-tillage with Stubble Mulching Enhances Soil Physical Properties and Reduces Soil Penetration Resistance: A Comparative Study in Mollisol Region of Northeast China"

_EGUsphere, 2025_

## Referee Comment (RC1)

Comparative experiments were as designed and conducted to discover effects of different mechanized tillage and sowing modes in mollisols region of Northeast China, and the soil physical characteristics of different models were systematically compared and analyzed according to experiments data. It shows that the experiment test scheme was reasonable, experiment the data was sufficient, the research result of the paper analysis was objective, the research methods were as proper to the contents of the paper, and the content is substantial of this article. And this article would be the powerful provided reference for the selection of appropriate mechanized tillage and sowing techniques in Northeast China's mollisols region, which had important guiding significance and practical value for the construction of rational plow cultivate layers and the implementation of conservation tillage technologies. However, the manuscript needs further significant revision to ensure scientific validity.

- 1. The title of the paper is "Enhances Soil Physical Properties and Reduces Soil Penetration Resistance...", but the paper only covers three indicators: soil solidity, soil moisture content, and soil bulk density, which seems inappropriate. Suggest modifying the title of the paper.
- 2. The four practices of handling expressions in the paper seem inaccurate.
- 3. The abstract does not provide a quantitative description of the indicators.
- 4. The paper describes the mechanized tillage and broadcasting mode, but the first keyword is "mechanized tillage practices". It is suggested to modify it to "mechanized tillage and sowing practices"
- 5. Suggest adding a review of the current situation both domestically and internationally in the introduction.
- 6. Pay attention to the uniformity of specialized terminology, such as correcting BD to SBD in line 50, and search for other similar issues on your own.
- 7. In section 2.1 Experimental sites, it is recommended not to include specific years for the description of annual rainfall and accumulated temperature in the experimental area; The experimental time and testing phase should appear in the 2.2 experimental design, and specific testing phases should be provided.
- 8. The experimental design department did not specify the rotation mode, nor did they provide specific testing data for which year in the rotation system; The description of the experimental treatment is not accurate enough. PTS, RTS, and NTS all require straw crushing treatment, which is also a key operational step to distinguish between NTS and NTMS, but it is not reflected in the text; The experimental equipment is also not clearly reflected here. wait.
- 9. The paper lacks a section on 'Statistical analysis'.
- 10. It is recommended not to use subheadings for patterns and stages in the description of each indicator in the Results section. Combining the 3-level headings of each indicator seems to facilitate understanding.
- 11. Separate the indicators and units in the title bar using () or/in the figure.
- 12. The conclusion summary is insufficient, highlighting important findings in the research.
- 13. Pay attention to the details, as the statement 'can 28.9% claim' on line 80 seems to be incorrect.
- 14. In addition, the presentation of the English language needs to be revised; the current manuscript has some colloquialisms. The authors are advised to make professional language polishing.